# Generalizability challenges of mortality risk prediction models: A retrospective analysis on a multi-center database

**Harvineet Singh**[1]*, **Vishwali Mhasawade**[2], **Rumi Chunara**[2,3]

**1** New York University, Center for Data Science, **2** New York University, Tandon School of Engineering, **3** New York University, School of Global Public Health

* hs3673@nyu.edu

## Abstract

Modern predictive models require large amounts of data for training and evaluation, absence of which may result in models that are specific to certain locations, populations in them and clinical practices. Yet, best practices for clinical risk prediction models have not yet considered such challenges to generalizability. Here we ask whether population- and group-level performance of mortality prediction models vary significantly when applied to hospitals or geographies different from the ones in which they are developed. Further, what characteristics of the datasets explain the performance variation? In this multi-center cross-sectional study, we analyzed electronic health records from 179 hospitals across the US with 70,126 hospitalizations from 2014 to 2015. Generalization gap, defined as difference between model performance metrics across hospitals, is computed for area under the receiver operating characteristic curve (AUC) and calibration slope. To assess model performance by the race variable, we report differences in false negative rates across groups. Data were also analyzed using a causal discovery algorithm "Fast Causal Inference" that infers paths of causal influence while identifying potential influences associated with unmeasured variables. When transferring models across hospitals, AUC at the test hospital ranged from 0.777 to 0.832 (1st-3rd quartile or IQR; median 0.801); calibration slope from 0.725 to 0.983 (IQR; median 0.853); and disparity in false negative rates from 0.046 to 0.168 (IQR; median 0.092). Distribution of all variable types (demography, vitals, and labs) differed significantly across hospitals and regions. The race variable also mediated differences in the relationship between clinical variables and mortality, by hospital/region. In conclusion, group-level performance should be assessed during generalizability checks to identify potential harms to the groups. Moreover, for developing methods to improve model performance in new environments, a better understanding and documentation of provenance of data and health processes are needed to identify and mitigate sources of variation.

**Data Availability Statement:** All data used in the study is publicly accessible from PhysioNet website https://physionet.org/content/eicu-crd/2.0/

upon completion of a training course and approval of data use request.

**Funding:** This study was funded by the National Science Foundation grant numbers 1845487 and 1922658. The funder had no role in design, conduct of the study or in preparing the manuscript.

**Competing interests:** The authors have declared that no competing interests exist.

## Author summary

With the growing use of predictive models in clinical care, it is imperative to assess failure modes of predictive models across regions and different populations. In this retrospective cross-sectional study based on a multi-center critical care database, we find that mortality risk prediction models developed in one hospital or geographic region exhibited lack of generalizability to different hospitals or regions. Moreover, distribution of clinical (vitals, labs and surgery) variables significantly varied across hospitals and regions. Based on a causal discovery analysis, we postulate that lack of generalizability results from dataset shifts in race and clinical variables across hospitals or regions. Further, we find that the race variable commonly mediated changes in clinical variable shifts. Findings demonstrate evidence that predictive models can exhibit disparities in performance across racial groups even while performing well in terms of average population-wide metrics. Therefore, assessment of sub-group-level performance should be recommended as part of model evaluation guidelines. Beyond algorithmic fairness metrics, an understanding of data generating processes for sub-groups is needed to identify and mitigate sources of variation, and to decide whether to use a risk prediction model in new environments.

## Introduction

Validation of predictive models on intended populations is a critical prerequisite to their application in making individual-level care decisions since a miscalibrated or inaccurate model may lead to patient harm or waste limited care resources [1]. Models can be validated either on the same population as used in the development cohort, named internal validity, or on a different yet related population, named external validity or generalizability (or sometimes transportability) [2]. The TRIPOD (Transparent Reporting of a multivariable prediction model for Individual Prognosis Or Diagnosis) Statement strongly recommends assessing external validity of published predictive models in multiple ways including testing on data from a different geography, demography, time period, or practice setting [3]. However, the guidelines do not specify appropriate external validity parameters on any of the above factors. At the same time, recent studies in the computer science and biomedical informatics literature have indicated that sub-group performance of clinical risk prediction models by race or sex can vary dramatically [4,5], and clinical behavior can guide predictive performance [6]. Within the statistics literature are several methods for computing minimum sample size and other best practices for assessing external validity of clinical risk prediction models [7–9]. However, the assessment of sub-group-level performance and data-shifts are not explicitly considered in such guidance [10]. Moreover, recent analyses have shown that clinical prediction models are largely being developed in a limited set of geographies, bringing significant concern regarding generalizability of models to broader patient populations [11]. Amidst such rising challenges, an understanding of how differences among population and clinical data impact external generalizability of clinical risk prediction models is imperative.

When a model fails to generalize for specific patient groups (such as racial or gender identities), using it to guide clinical decisions can lead to disparate impact on such groups. This raises questions of equity and fairness in the use of clinical risk prediction models, for which performance on diverse groups has been repeatedly lacking [12–14]. Predictive discrimination quantifies how well a model can separate individuals with and without the outcome of interest (we study mortality prediction here). Calibration quantifies how well the predicted probabilities match with the observed outcomes. These measures can be used to check for aggregate

model performance across a study sample or within groups, but do not illuminate variation across groups. Hence, in assessment of generalizability, we add another set of measures to our analyses which we refer to as "fairness" metrics, following the algorithmic fairness literature [15–17]. Such performance checks are important, especially given the evidence on racial bias in medical decision-making tools [4,13,14].

The primary objective of this study is to evaluate the external validity of predictive models for clinical decision making across hospitals and geographies in terms of the metrics–predictive discrimination (area under the receiver operating characteristic curve), calibration (calibration slope) [18], and algorithmic fairness (disparity in false negative rates and disparity in calibration slopes). The secondary objective is to examine the possible reasons for performance changes via shifts in the distributions of different types of variables and their interactions. We focus on risk prediction models for in-hospital mortality in ICUs. Our choice of evaluation metrics are guided by the use of such models for making patient-level care decisions. We note that similar models (e.g., SAPS and APACHE scores) [19,20] are widely-used for other applications as well such as assessing quality-of-care, resource utilization, or risk-adjustment for estimating healthcare costs [19–21], which are not the focus of this study. Recently, prediction models for in-hospital mortality have been prospectively validated for potential use [22], or in case of sepsis, have even been integrated into the clinical workflow [23]. With access to large datasets through electronic health records, new risk prediction models leveraging machine learning approaches have been proposed, which provide considerable accuracy gains [24]. Being flexible, such approaches might overfit to the patterns in a particular dataset, thus, raising concerns for their generalization to newer environments [25]. We use the eICU dataset [26] as a test bed for our analyses. Past studies have employed the dataset for evaluating mortality prediction models [27,28]. As the dataset was collected from multiple hospitals across the US, it allows us, in a limited way, to test external validity across hospitals, diverse geographies, and populations.

## Materials and methods

Analyses are based on data obtained from the publicly-available eICU Collaborative Research Database [26], designed to aid remote care of critically-ill patients in a telehealth ICU program. The database is composed of a stratified random sample of ICU stays from hospitals in the telehealth program where the sample is selected such that the distribution of the number of unique patient-stays across hospitals is maintained [26]. Data on 200,859 distinct ICU stays of 139,367 patients with multiple visits across 208 hospitals in the US between 2014 and 2015 are included. We followed the Strengthening the Reporting of Observational Studies in Epidemiology (STROBE) reporting guideline [29].

### Data preprocessing

We follow the feature extraction and exclusion procedures including the exclusion criteria used by Johnson et al [27] which removes patient stays conforming to APACHE IV exclusion criteria [20] and removes all non-ICU stays. APACHE IV criteria excludes patients admitted for burns, in-hospital readmissions, patients without a recorded diagnosis after 24 hours of ICU admission, and some transplant patients [26]. Only patients 16 years or above are included. Age of patients above 89 years (which is obfuscated to adhere to HIPAA provisions) is coded as 90. After pre-processing, the dataset consists of 70,126 stays from 179 hospitals. For analyses, data is grouped at two levels–by individual hospitals and by U.S. geographic regions (Northeast, South, Midwest, West) [30]. Hospital-level analyses are restricted to the top 10 hospitals with the most stays, all of which have at least 1631 stays, to ensure enough examples

for model training and evaluation. Data is split into ten separate datasets using a hospital identifier for hospital-specific analyses and into four separate datasets using a region identifier for region-specific analyses. The outcome label is in-hospital mortality (binary). Mortality rates differed in the range of 3.9%-9.3% (1st-3rd quartile) across hospitals. Summary statistics by hospital and region are included in Table A and Table B in S1 Text.

## Mortality prediction model

Features from the SAPS II risk scoring model [19] from the first 24 hours of the patient-stay starting from ICU admission were extracted and are summarized in Table C in S1 Text. These include 12 physiological measurements (vitals and labs), age, and an indicator for whether the stay was for an elective surgery. For features with multiple measurements, their worst values determined using the SAPS II scoring sheet (Table 3 in Le Gall et al [19]) are extracted. For example, for Glasgow Coma Score, we take the minimum value among the measurements. As previously employed for mortality prediction [27], we use logistic regression with $\ell_2$ regularization using the implementation in scikit-learn v0.22.2 package with default hyperparameters [31]. Missing values in features are imputed with mean values computed across the corresponding columns of the full dataset. We experimented with other imputation methods as well, specifically imputation with mean or median across the train datasets and single imputation with a decision tree [32], however, the conclusions did not change. Features are then standardized to zero mean and unit variance using statistics from the train datasets. As sample size used in training models can affect generalizability, we control for this factor by fixing the number of samples used for training and testing. We use 1631 (or 5000) samples from each hospital (or region) while training and testing models. For model development, each dataset (for a hospital or region) is randomly split with the training set comprising 90% of samples and the validation set comprising the remaining 10%. The test set comprises all samples from the hospital (or region) different from the one included in the training set.

## Statistical analysis

**Performance metrics.** Discrimination ability of the models is assessed using area under the receiver operating characteristic curve (AUC). For binary outcomes, calibration slope (CS) is computed as the slope of the regression fit between true outcomes and logits of the predicted mortality, with a logit link function. A perfectly calibrated model has a CS of 1. A value lower than 1 indicates that the risk estimates are extreme, i.e. overestimation for high risk patients and underestimation for low risk patients, and suggests overfitting of the model [33]. Hence, a value close to 1 is desirable. In addition to reporting AUC and CS computed on the test sets, we also report how much the metrics differ from their values computed on the validation set. This difference, known as the generalization gap, provides a quantitative measure of generalization performance (e.g. Jiang et al [34]). If this difference is high, i.e. the test metrics are worse than the train metrics, the model is said to lack generalizability. The allowable difference between test and train depends on the application context. Studies typically report confidence intervals around the difference and/or the percentage change relative to the train performance [35]. To measure fairness of model predictions, we use the racial/ethnic attributes to form two groups–African American, Hispanic, and Asian as one group and the rest as another. These will be referred to as *minority* and *majority* groups. Note that the racial/ethnic attributes are used only for the purposes of fairness analysis, these are not part of the model building process. We acknowledge that aggregating multiple groups does not represent an exposition of which groups are advantaged in the models. Based on the available dataset sizes, this approach serves to illustrate differences that would ideally be unpacked in detail in future work addressing

such issues. Disparity in false negative rates (DisparityFNR), and disparity in calibration slope (DisparityCS) are computed as the difference between the respective metric's value for the minority and the majority group. Differences in these two metrics have been employed in recent studies for bias analysis [16,17]. FNR quantifies the rate at which patients with the observed outcome of death were misclassified. Thus, a high FNR for the score may lead to an increase in undertreatment, and high DisparityFNR (in absolute value) highlights large differences in such undertreatment across groups. For the prediction threshold for FNR we use the mortality rate at the *test* hospital (assuming it is known beforehand). This threshold can be chosen in a more principled way, for example, based on decision-curve analysis [18], which will depend on the application context. We further acknowledge that there are myriad ways to define fairness that will depend on the context of the risk prediction's use and inputs from stakeholders [36].

**Dataset differences.**   To address our secondary objective of studying external validity-specific performance changes, we test for dataset shifts across hospitals and geographies (i.e. whether the distributions of two datasets differ), and use causal graph discovery to explore the reasons for these differences. Dataset shifts are measured using squared maximum mean discrepancy ($MMD^2$) [37]. We perform the two-sample tests under the null hypothesis that the distributions are the same and threshold the resulting p-values at the significance level of 0.05. Details of the $MMD^2$ metric and the hypothesis test are included in Method A in S1 Text. To explain the shifts we leverage the recently introduced framework of Joint Causal Inference [38] which allows constructing a single graphical representation of how variables relate to each other, in the form of a causal graph. We use the Fast Causal Inference (FCI) algorithm [39] for constructing the causal graph as it is methodologically well-developed and requires fewer assumptions on the data generating process as it allows for the presence of unobserved variables affecting the observed variables in the data (i.e. unobserved confounders). We also include race as an indicator in the datasets, before running FCI, to study the causes of unfairness with respect to the race variable. Details of the causal modeling are included in Method B in S1 Text. Note that we use causal graphs as a compact representation of (conditional) independencies in the datasets. These are not meant to make statements about the causal effect of treatments on physiological variables, for which randomized controlled trials and other methods may be used to gather better evidence.

The metrics of interest–AUC, CS, DisparityFNR, DisparityCS, p-value, and $MMD^2$ –are averaged over 100 random subsamples of the datasets (i.e. resampling without replacement). While aggregating across hospitals (or regions), we report the median, 1st, and 3rd quartiles across all train-test set pairs which includes the 100 random subsamples in each pair.

## Results

Fig 1 demonstrates the highly varied external validity of models across hospitals based on AUC, CS, generalization gap in AUC and in CS, DisparityFNR, and DisparityCS. Across all train-test hospital pairs, median AUC is 0.801 with 1st-3rd quartile range (IQR) as 0.778 to 0.832, and CS is 0.853 (IQR 0.725 to 0.983). AUCs are lower than the typical values for mortality risk prediction models of around 0.86 [19,22], although AUCs in the same range (around 0.8) have been observed in other studies (albeit in different populations) and were considered acceptable [40–42]. CS of around 0.8, as observed in our case, is considered to indicate overfitting [7]. Transferring a model trained on hospital ID 73, which is the hospital with the most samples, to other hospitals results in a median gap in AUC of -0.087 (IQR -0.134 to -0.046) and a median gap in CS of -0.312 (IQR -0.502 to -0.128). In aggregate, we observe a decline in the performance on the test hospitals relative to that on the train hospitals (Fig 1A). Across all

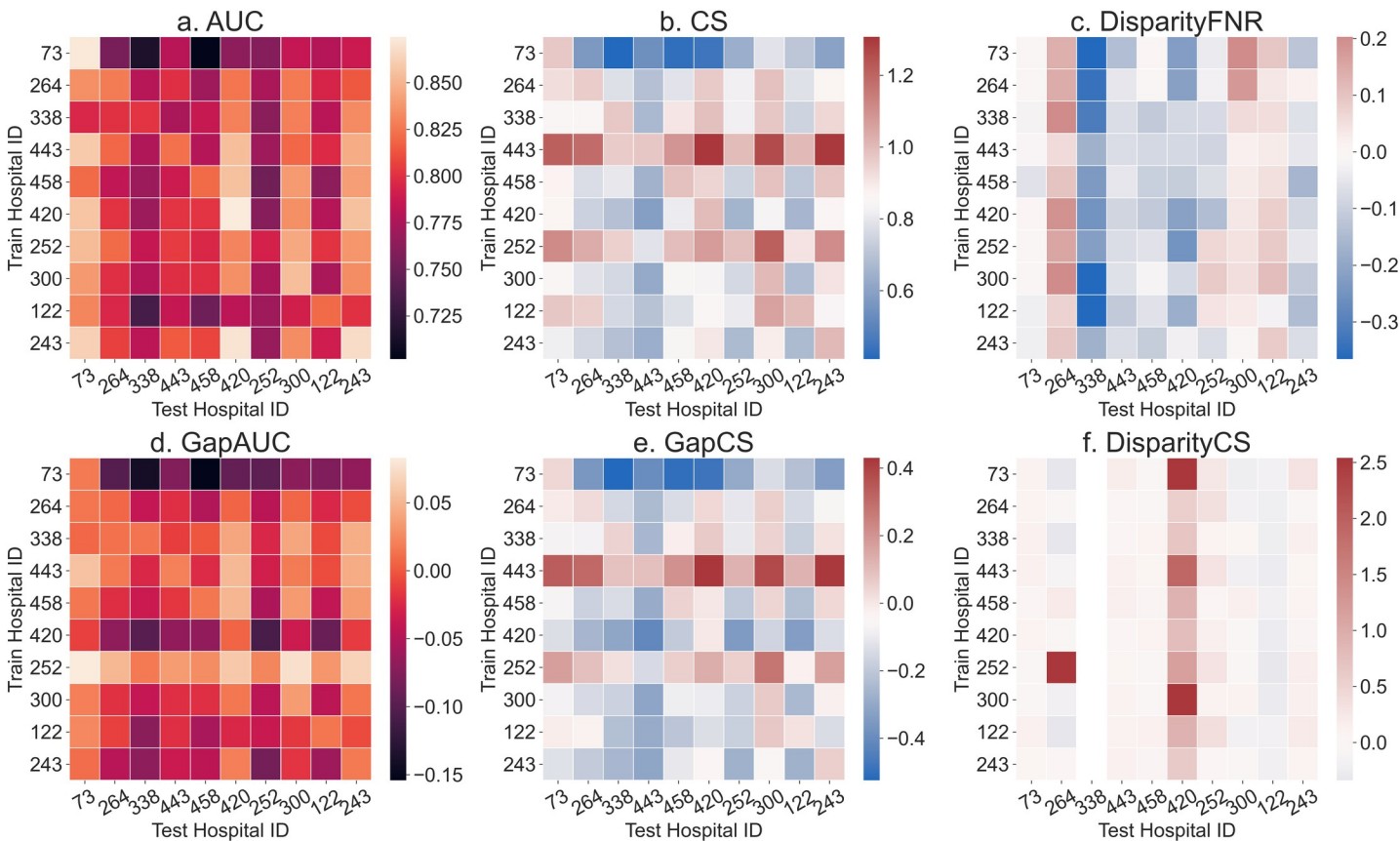

**Fig 1. Generalization of performance metrics across individual hospitals.** Results of transferring models across top 10 hospitals by number of stays. Models are trained and tested on a fixed number of samples (1631, the least in any of the 10 hospitals) from each hospital. Results are averaged over 100 random subsamples for each of the 10×10 train-test hospital pairs. All 6 metrics show large variability when transferring models across hospitals. Abbreviations: AUC, area under ROC curve; CS, calibration slope; FNR, false negative rate.

train-test hospital pairs, the median generalization gap in AUC is -0.018 (IQR -0.065 to 0.032) and the median generalization gap in CS is -0.074 (IQR -0.279 to 0.121). Fig 1B shows that the majority of models have CS of less than 1, indicating consistent miscalibration of mortality risk at test hospitals. This conforms with the typical observation of good discriminative power but poor calibration of SAPS II models [40,42–44]. The median values of AUC and CS are negative, indicating that both of them decrease in majority of the cases upon transfer. For comparison, the generalization gap in AUC for the SAPS II score in the original study by Le Gall et al [19] was -0.02 (AUC decreased to 0.86 in validation from 0.88 in training data), which is the same as the median gap here. Thus, for more than half of the hospital pairs the AUC drop is worse than the acceptable amount found in the original SAPS II study. Percentage changes in AUC and CS from train to test set, reported in Table D in S1 Text also indicate substantial drop in performance (in the range of -2.5% to -31.5% in AUC and -15.9% to -45.4% in CS). In some cases, for example for hospital ID 252, we observe an improvement in AUC (fourth row from bottom, Fig 1D). With regard to fairness metrics, DisparityFNR (absolute value) has median 0.093 (IQR 0.046 to 0.168), i.e. false negative rates across the racial groups differ by 4.6% to 16.8%. DisparityCS, i.e. the absolute value of difference in calibration across racial groups, is large as well (median 0.159; IQR 0.076 to 0.293). Considering that the ideal DisparityCS is 0, when CS is 1 for both the groups, the observed DisparityCS of 0.159 is large. There

are both positive and negative values in the disparity metrics (Fig 1C and 1F), i.e. models are unfair to the minority groups for some pairs and vice versa for others. Note that disparity metrics for hospital ID 338 are considerably different from others (third column in Fig 1C and 1F) due to the skewed race distribution with only 74 (3.2%) samples from the minority groups (Table A in S1 Text). We observe that the variation across hospitals in fairness metrics (DisparityFNR, and DisparityCS) is not captured by the variation in discrimination and calibration metrics (AUC and CS). Thus, fairness properties of the models are not elucidated by the standard metrics and should be audited separately.

Next, given concerns about the development of machine learning models in a limited set of geographies [11], we pool hospitals by geographic region, and validate models trained in one region and tested on another (Fig 2). Performance in terms of AUC and CS across regions improves as a result of pooling hospital data. Overall, AUC varies in a small range (median 0.804; IQR 0.795 to 0.813) as does CS (median 0.968; IQR 0.904 to 1.018). The same can be observed through generalization gaps in AUC and CS which are smaller–median generalization gap in AUC is -0.001 (IQR -0.017 to 0.016) and median generalization gap in CS is -0.008 (IQR -0.081 to 0.075). However, such pooling does not alleviate fairness metric disparities. DisparityFNR (absolute value) has a median value of 0.040 (IQR 0.018 to 0.074). This translates to, for example, a disparity between minority and majority groups of 6.36% (95% CI

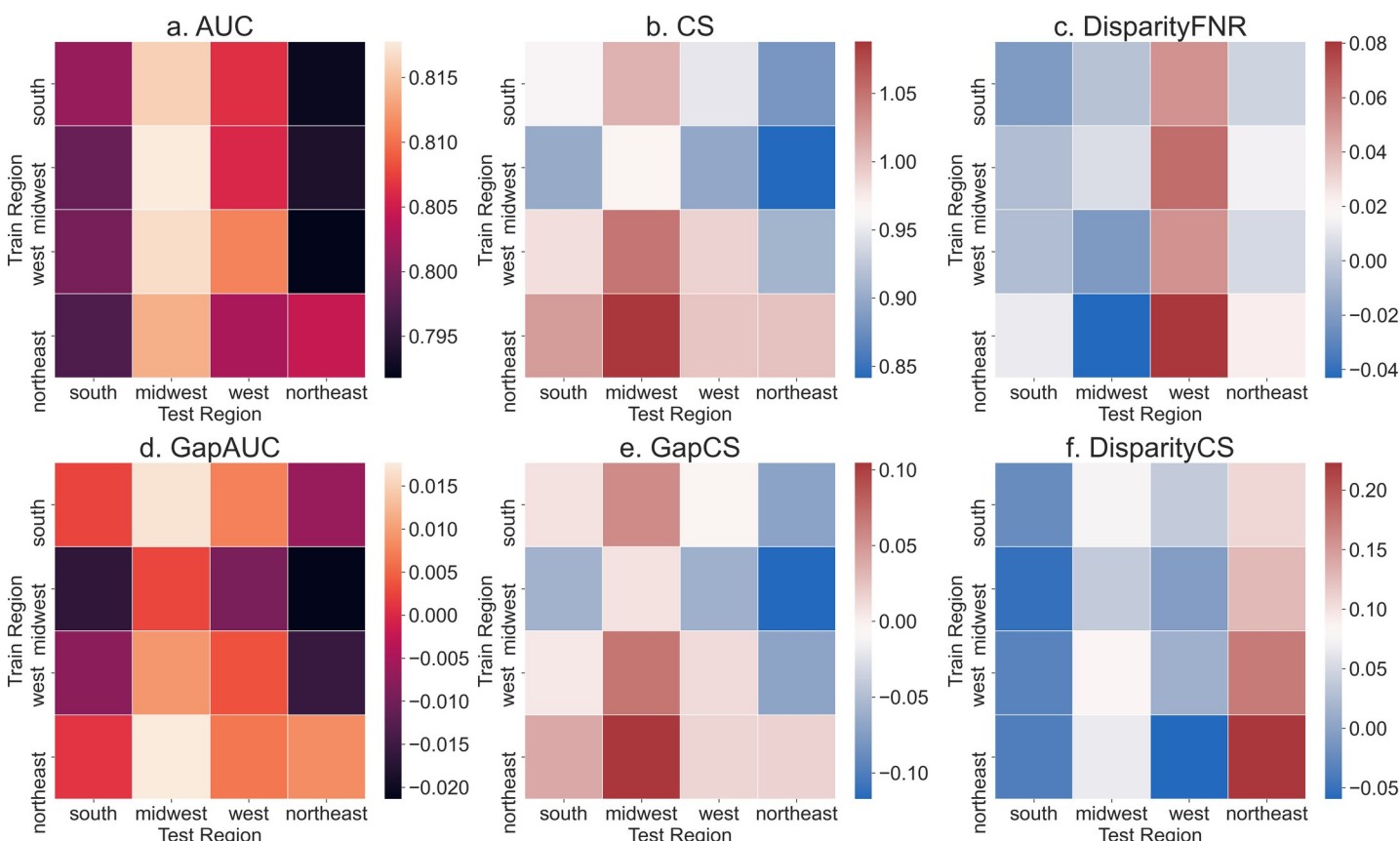

**Fig 2. Generalization of performance metrics across US geographic regions.** Results of transferring models after pooling hospitals into 4 regions (northeast, south, midwest, west). Models are trained and tested on 5000 samples from each region. Results are averaged over 100 random subsamples for each of the 4×4 train-test hospital pairs. DisparityFNR and DisparityCS show large variability when transferring models across regions. Abbreviations: AUC, area under ROC curve; CS, calibration slope; FNR, false negative rate.

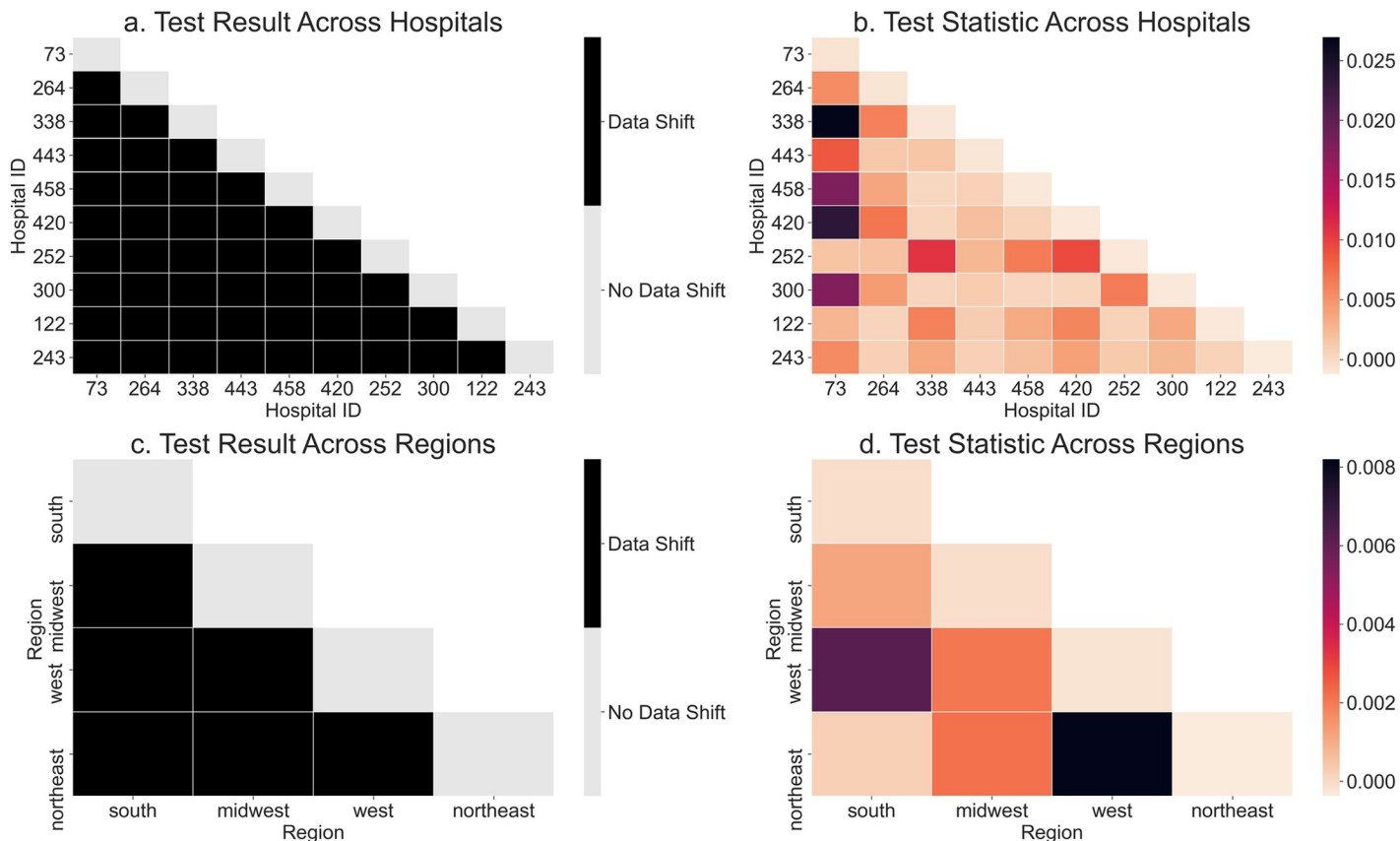

**Fig 3. Statistical tests for dataset shifts.** Results for two-sample tests with and without pooling of hospitals by region. Test results are plotted in (a,c) and test statistics are plotted in (b,d) to examine the test results in more detail. Since the order of hospitals considered in the two-sample test does not change the test statistic, we plot only the lower halves of the matrices. Results are averaged over 100 random subsamples. Feature distribution changes across all hospital and region pairs.

-7.66% to 17.42%) and 8.06% (95% CI -3.81% to 19.74%) when transferring models from Midwest to West and Northeast to West respectively (though CIs are large, both values are greater than 0%; one-sided one-sample t-test, p = $10^{-5}$). DisparityCS (absolute value) is still high with a median value of 0.104 (IQR 0.050 to 0.167). For example, transferring models from South to Northeast (the region with the least minority population size) has a high DisparityCS (median 0.108; IQR -0.015 to 0.216). Percentage change in the test set metrics relative to the train set is reported in Table E in S1 Text which shows significant changes in DisparityFNR (ranging from -33% to 65%) and DisparityCS (ranging from -52% to 67%). Apart from geography, differences across hospitals can also be due to differences in patient load and available resources. In Figure B in S1 Text we include results for more fine-grained pooling of hospitals based on their number of beds, teaching status, and region where we again find consistent lack of generalizability in fairness metrics.

To investigate reasons for these performance differences across hospitals and regions, we first consider whether the corresponding datasets differ systematically. Fig 3 shows results from statistical tests for dataset shifts across hospitals and regions. Shifts across all pairs of hospitals are significant. Some hospitals are considerably different from others like hospital ID 73 in the first column of Fig 3B, which has significantly lower mortality rate than the other hospitals (Table A in S1 Text).

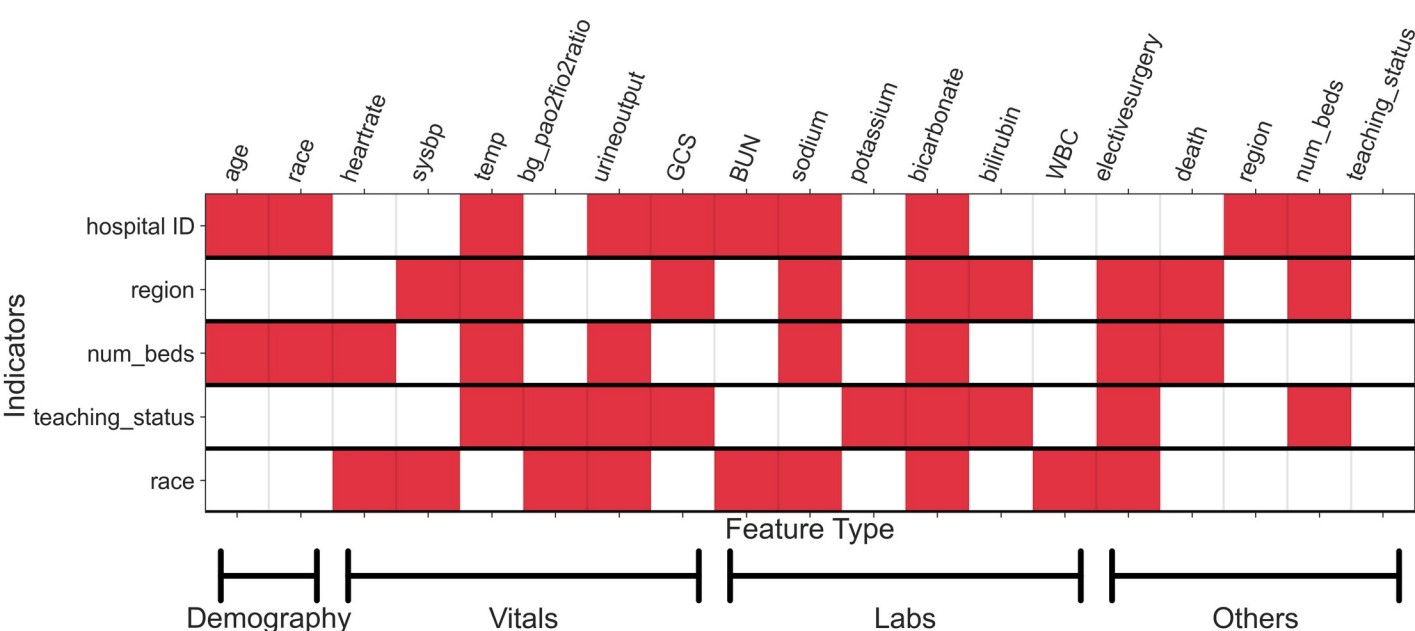

**Fig 4. Shifts in variable distributions due to hospital, region, and other factors based on mortality causal graph.** Each row represents (in red) the features which explain the shifts across each of the indicators labeling the row, i.e. the features with an edge from the indicator in the causal graph. For instance, shift across the hospital ID indicator (first row) is explained by shifts in the distributions of age, race, temp (or temperature), urine output, and so on. We observe that shifts are explained by changes in a few variables which are common across indicators. Full forms of the abbreviated feature names are added in Table C in S1 Text.

Finally, we study explanations for the observed shifts in Fig 3A via individual features. The discovered causal graph is included in Figure C in S1 Text which shows the estimated causal relationships among clinical variables, and which of these variables shift in distribution based on hospital, geography and other factors (i.e. which variables have a direct arrow from the indicators like hospital or region). Fig 4 summarizes the shifts from the causal graph. From Fig 4, we note that the distribution of all fourteen features and the outcome are affected either directly or indirectly by the hospital indicator. However, restricting to direct effects of hospital (first row in Fig 4), we observe that shifts are explained by few of the features–demography (age and race), vitals (3 out of 6), and labs (3 out of 6). Not all vitals and labs change directly as a result of a change in hospitals; changes in 3 of the vitals and 3 of the labs are mediated through changes in other features. We attempt to further understand these changes across hospitals by including hospital-level contextual information, namely, their region, size (number of beds), and teaching status. We observe that there exists common features that explain shifts across the three attributes (different rows in Fig 4). Thus, some of the variation across hospitals is explained through its region, size, and teaching status. But, notably, the three attributes do not explain all variation among hospitals and more contextual information is required. For the fairness analysis, we observe the direct effects of the race variable (last row in Fig 4). There are direct effects from the race variable to most vitals (4 out of 6), labs (4 out of 6), and indicator for elective surgery. These direct effects support past observations made on racial disparities, e.g. in access to specialized care (race → elective surgery) [45] and in blood pressure measurements (race → sysbp or systolic blood pressure) [46]. Out of the 9 features that are directly affected by the race variable, 4 also vary across the hospitals. This suggests that the distribution of clinical variables for the racial groups differs as we go from one hospital to another. As a result, the feature-outcome relationships learnt in one hospital will not be suitable in another,

leading to the observed differences in model performance (as quantified by the fairness metrics) upon model transfer.

## Discussion

We retrospectively evaluate generalizability of mortality risk prediction models using data from 179 hospitals in the eICU dataset. In addition to commonly-used metrics for predictive accuracy and calibration, we assessed generalization in terms of algorithmic fairness metrics. To interpret results, we investigated shifts in the distribution of variables of different types across hospitals and geographic regions, leading to changes in the metrics. Findings highlight that recommended measures for checking generalizability are needed, and evaluation guidelines should explicitly call for the assessment of model performance by sub-groups.

Generalizability of a risk prediction model is an important criteria to establish reliable and safe use of the model even under care settings different from the development cohort. A number of studies have reported lack of generalizability of risk prediction models across settings such as different countries [47–50], hospitals [27,51–54], or time periods [55–57]. For instance, Austin et al [58] study the validity of mortality risk prediction models across geographies in terms of discrimination and calibration measures. They find moderate generalizability, however, the hospitals considered belonged to a single province in Canada. More importantly, these works did not investigate generalizability for different groups in the patient cohorts. Results in Fig 1 show that the fairness characteristics of the models can vary substantially across hospitals. Prior work investigating algorithmic fairness metrics in a clinical readmission task [4] did not investigate changes in the metric when models are transferred across care settings. A recent study of a mortality prediction model showed good performance across 3 hospitals (1 academic and 2 community-based) as well as good performance for subgroups *within* a hospital [22]. However, the change in performance for the subgroups *across* hospitals was not explored.

One strategy to tackle the lack of generalizability is to pool multiple hospital databases to potentially increase diversity of the data used in modeling [59]. However, available databases may not faithfully represent the intended populations for the models even after pooling. A recent study [11] found that US-based patient cohorts used to train machine learning models for image diagnosis were concentrated in only three states. However, the effect of using geographically-similar data on generalizability and fairness had not been studied previously. Results in Fig 2 suggest that pooling data from similar geographies may not help mitigate differences in model performance when transferred to other geographies. This finding adds more weight to the concerns raised about possible performance drop when transferring models from data-rich settings to low-resource settings [60].

Prior work has postulated multiple reasons for lack of generalizability [61] including population differences and ICU admission policy changes [47]. In Davis et al [55], reasons including case mix, event rate, and outcome-feature association are assessed. However, specific variables which shift across clinical datasets have not been examined. Through an analysis of the underlying causal graph summarized in Fig 4, we identify specific features that explain the changes in data distributions across hospitals. Demographics (age and race) differ across hospitals which is aligned with population difference being the common reason cited for lack of generalizability [47,55]. We find that vitals and labs differ as well but only a few change as a direct consequence of changes in the hospital setting. Often, the changes are mediated via a small number of specific vitals and labs. For understanding the root causes of the shifts, causal graph analysis helps to narrow down candidate features to analyse further. Significant differences found across ICUs in feature distributions and model performance calls for a systematic

approach to transferring models. Understanding the reasons for the lack of generalizability is the first step to deciding whether to transfer a model, re-training it for better transfer, or developing methods which can improve transfer of models across environments [62]. Factors affecting generalizability can potentially be due to variations in care practices or may represent spurious correlations learned by the model. These findings reinforce calls to better catalog clinical measurement practices [63] and continuously monitor models for possible generalizability challenges [61].

Beyond the descriptive analyses of dataset shifts with causal graphs, in Figure A in S1 Text, we examine whether the shifts can *predict* lack of generalizability. We plot the generalization gap in AUC and CS against the amount of shift, as measured in $MMD^2$, and report the correlation coefficient. Although we find only low correlation, this suggests a need to develop better methods of quantifying dataset shifts that can predict future model performance. One such metric derived from a model trained to discriminate between training and test samples was found to explain the test performance well (focusing on environments that primarily differ by case-mix, without attention to specific clinical or demographic shifts) [64]. We hope that current work motivates development and evaluation of such metrics on larger and more diverse populations and datasets. Recent work has also proposed methodological advances to ensure transferability of models across settings, for example, by pre-training on large datasets from related machine learning tasks [65,66] and with the help of causal knowledge about shifts [67–69]. Better metrics for dataset shift can help practitioners decide whether to transfer a model to a new setting based on how large the shift is between hospitals, for example.

Importantly, findings here showed that the race variable often mediated shifts in clinical variables. Reasons for this must be disentangled. As race is often a proxy variable for structural social processes such as racism which can manifest both through different health risk factors as well as different care (differences in health care received by patients' racial group are well-documented) [70–73], shifts across hospitals cannot be mitigated simply by population stratification or algorithmic fairness metrics alone. Indeed, better provenance of the process by which data is generated will be critical in order to disentangle the source of dataset differences (for example, if clinical practices or environmental and social factors are giving rise to different healthcare measures and outcomes). Following guidelines developed for documenting datasets [74] and models [75] in the machine learning community, similar guidelines should be established for models in healthcare as well [10]. An example is the proposal for reporting sub-group-level performances in MI-CLAIM checklist [76]. Further, the datasets have a skewed proportion of the minority population (as low as 3.2% for hospital ID 338, Table A in S1 Text). This may negatively impact model performance for minority groups since the number of training samples might be insufficient to learn group-specific outcome characteristics, which is a recognized source of bias termed as representation bias [77]. To reduce the impact of skewed data, algorithmic fairness literature proposes strategies such as reweighting data points for different groups based on their proportions or classification error [78–80]. Efficacy of such strategies to decrease the disparity metrics reported in our results can be investigated in future work. In sum, our findings demonstrate that data provenance, as described above, is needed in addition to applying algorithmic fairness metrics alone, to understand the source of differences in healthcare metrics and outcomes (e.g. clinical practice versus other health determinants), and assess potential generalizability of models.

## Limitations

The main limitation of this study is that the results are reported for a collection of ICUs within the same electronic ICU program by a single provider. This collection captures only a part of

the diversity in care environments that a mortality prediction model might be deployed in. Further, our investigation is limited to models constructed using the SAPS II feature set containing 14 hand-crafted features for the mortality prediction task. Though we employ widely-used methods, our analysis is limited by the specific methods used for computing dataset differences, building predictive models and causal graphs. For analyzing reasons for dataset shifts, we could only investigate explanations based on the 14 features along with limited hospital characteristics (such as geographic region, number of beds, and teaching status). Multiple factors are left unrecorded in the eICU database, such as patient load, budget constraints, and socioeconomic environment of the hospital's target population, that may affect the care practices and outcomes recorded in the dataset. We do not investigate dataset shifts in time, which are common [81], as the eICU database includes patient records only for a year. Finally, we do not assess the effect of resampling data points on model performance to address issues of skewed mortality-class distribution or minority-majority group distribution.

## Conclusion

External evaluation of predictive models is important to ensure their responsible deployment in different care settings. Recommended metrics for performing such evaluation focus primarily on assessing predictive performance of the models while ignoring their potential impact on health equity. Using a large, publicly-available dataset of ICU stays from multiple hospital centers across the US, we show that models vary considerably in terms of their discriminative accuracy and calibration when validated across hospitals. Fairness of models, quantified using their differential performance on racial groups, is found to be lacking as well. Furthermore, fairness metrics continue to be poor when validating models across US geographies and hospital types. Importantly, the pattern of out-of-sample variation in the fairness metrics is not the same as that in the accuracy and calibration metrics. Thus, the standard checks do not give a comprehensive view of model performance on external datasets. This motivates the need to include fairness checks during external evaluation. While examining reasons for the lack of generalizability, we find that population demographics and clinical variables differ in their distribution across hospitals, and the race variable mediates some variation in clinical variables. Documentation of how data is generated within a hospital where a model is developed specific to sub-groups, along with development of metrics for dataset shift will be critical to anticipate where prediction models can be transferred in a trustworthy manner.

## Supporting information

**S1 Text. Method A.** Difference in Distributions and Statistical Tests. **Method B.** Causal Graph Discovery. **Method C.** Percentage Change. **Table A.** Summary statistics for data after grouping based on hospital ID. **Table B.** Summary statistics for data after grouping based on regions. **Table C.** List of features **Table D.** Percentage change in metrics from the train hospital to the rest. **Table E.** Percentage change in metrics from the train region to the rest. **Table F.** Description of edge types in causal graph. **Figure A.** Generalization gap in AUC and CS versus dataset shift. **Figure B.** Generalization of performance metrics across regions, number of beds, and teachingstatus. **Figure C.** Causal graph discovered with FCI algorithm.
(DOCX)

## Author Contributions

**Conceptualization:** Harvineet Singh, Vishwali Mhasawade, Rumi Chunara.

**Data curation:** Harvineet Singh, Vishwali Mhasawade.

**Formal analysis:** Harvineet Singh, Vishwali Mhasawade.

**Funding acquisition:** Rumi Chunara.

**Investigation:** Harvineet Singh, Vishwali Mhasawade.

**Methodology:** Harvineet Singh, Vishwali Mhasawade, Rumi Chunara.

**Project administration:** Rumi Chunara.

**Resources:** Rumi Chunara.

**Software:** Harvineet Singh, Vishwali Mhasawade.

**Supervision:** Rumi Chunara.

**Validation:** Harvineet Singh, Vishwali Mhasawade.

**Visualization:** Harvineet Singh, Vishwali Mhasawade.

**Writing – original draft:** Harvineet Singh, Vishwali Mhasawade, Rumi Chunara.

**Writing – review & editing:** Harvineet Singh, Vishwali Mhasawade, Rumi Chunara.

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
