## [Decision Letter · Decision Letter 0]

29 Sep 2021

PDIG-D-21-00052

Generalizability Challenges of Mortality Risk Prediction Models: A Retrospective Analysis on a Multi-center Database

PLOS Digital Health

Dear Dr. Singh,

Thank you for submitting your manuscript to PLOS Digital Health. After careful consideration, we feel that it has merit but does not fully meet PLOS Digital Health’s publication criteria as it currently stands. Therefore, we invite you to submit a revised version of the manuscript that addresses the points raised during the review process.

Thanks for your submission. Overall, the comments from our reviewers are very positive. The study tackles an interesting topic and it is well implemented and clearly described. Some issues are noted in the reviews, all of which appear relatively minor. I would be grateful if you could resubmit a revised version of the manuscript after responding to the points raised. Thanks!

We look forward to receiving your revised manuscript.

Kind regards,

Tom J. Pollard, Ph.D.

Academic Editor

PLOS Digital Health

Journal Requirements:

1. Please expand the acronym “NSF” (as indicated in your Financial Disclosure) so that it states the name of your funders in full.

2. Please provide separate figure files in .tif or .eps format only, and remove any figures embedded in your manuscript file. If you are using LaTeX, you do not need to remove embedded figures.

Additional Editor Comments (if provided):

Reviewers' comments:

Reviewer's Responses to Questions

**Comments to the Author**

1. Does this manuscript meet PLOS Digital Health’s publication criteria? Is the manuscript technically sound, and do the data support the conclusions? The manuscript must describe methodologically and ethically rigorous research with conclusions that are appropriately drawn based on the data presented.

Reviewer #1: Yes

Reviewer #2: Yes

Reviewer #3: Yes

2. Has the statistical analysis been performed appropriately and rigorously?

Reviewer #1: Yes

Reviewer #2: Yes

Reviewer #3: Yes

3. Have the authors made all data underlying the findings in their manuscript fully available (please refer to the Data Availability Statement at the start of the manuscript PDF file)?

Reviewer #1: Yes

Reviewer #2: Yes

Reviewer #3: Yes

4. Is the manuscript presented in an intelligible fashion and written in standard English?

Reviewer #1: Yes

Reviewer #2: Yes

Reviewer #3: Yes

5. Review Comments to the Author

Reviewer #1: This study explores the generalizability of training models on one set of data, from a particular hospital or region, and using that model for mortality risk prediction on a different hospital or region. The topic explored here is very relevant to the state of science in this field. I support the effort to develop a better understanding of what contributes to a lack of generalizability from training models in one area and applying them in another. This will help develop better models in the future. However, I do think there are some areas which need to be addressed.

1. Training models from a given region/hospital and then testing those models on a different hospital/region is certainly a good way to explore generalizability. However, it may also be interesting to train on all of the data and then check to see how that generalizes to specific hospitals/regions. Was this approach considered as part of the study design?

2. In the S1 Table, for the electivesurgery variable, in nine of the ten hospitals false is much more common than true but for Hospital ID 458 the opposite it true. Is the result shown for this hospital correct?

3. In the S2 Table, the median age of patients is exactly the same across all four regions. Is this correct?

4. On line #245, the text mentions changes in AUC and CS but numerical results are only provided for AUC. I’d recommend showing the numerical result for CS also since it is mentioned in the text alongside AUC.

5. On line #290, the text references Figure S3 but does not explain what it is showing. If a figure is referenced in the results section it should be accompanied by a brief explanation about what it is showing.

6. On line #308, the term “high-quality care” is used when discussing the direct effects of race on elective surgery. It isn’t clear to me that quality of care is directly related to whether or not elective surgery is chosen. Perhaps this should be explained further or another term should be used for the effect of race on elective surgery.

7. Finally, I’d encourage double-checking the manuscript for grammatical errors. I noticed a number of sentences which were missing words.

Reviewer #2: In this paper, a multi-center cross-sectional study was performed using data electronic health records from eICU database to predict mortality risk, considering the generalizability challenge. A causal discovery algorithm achieved good AUC performance. Other performance metrics which are useful to analyze modeling results were reported. The authors found that the distribution of demographic, vitals and laboratory variables differed significantly across hospitals and regions and that Race mediated differences in the relationship between clinical variables and mortality, by hospital/region.

The paper is in general well written and the findings are interesting.

I only have minor comments.

In the introduction, the sentence in line 73 should be re-written for a clear understanding.

Did the authors consider other missing data imputation techniques for the model covariates?

The values of the covariates in the training dataset should be used as reference to apply the imputation mean in the test set, otherwise results might be biased. The authors mention that the mean is applied to data, in line 153, but do not specify this detail. The difference in mean and standard deviation between the train and the full datasets might be small and not cause an impact on the normalization results, but this should be checked.

In line 166 correct “a CS”.

In the paragraph correspondent to line 216, the authors seem to be describing “bootstrapping”.

When referring to the variables such as the example "urineoutput" in the main text, the authors may write "urine output".

Reviewer #3: Excellent work. The analysis of models scoring metrics is well presented. The research clearly highlights the current imbalance between achieveing ever-higher AUCs despite negletcing overfitting and the generalization ability on different cohorts. The use of causal inference estimation algorithms supports the hypothesis proposed by the authors while helping the reader further grasping the nature of the problem.

6. PLOS authors have the option to publish the peer review history of their article (what does this mean?). If published, this will include your full peer review and any attached files.

**Do you want your identity to be public for this peer review?** For information about this choice, including consent withdrawal, please see our Privacy Policy.

Reviewer #1: No

Reviewer #2: No

Reviewer #3: No

---

## [Editor Report · Decision Letter 1]

18 Jan 2022

PDIG-D-21-00052R1

Generalizability Challenges of Mortality Risk Prediction Models: A Retrospective Analysis on a Multi-center Database

PLOS Digital Health

Dear Dr. Singh,

Thank you for submitting your manuscript to PLOS Digital Health. After careful consideration, we feel that it has merit but does not fully meet PLOS Digital Health's publication criteria as it currently stands. Therefore, we invite you to submit a revised version of the manuscript that addresses the points raised during the review process.

We look forward to receiving your revised manuscript.

Kind regards,

Harry Hochheiser

Section Editor

PLOS Digital Health

Journal Requirements:

1. Please update your Competing Interests statement. If you have no competing interests to declare, please state: “The authors have declared that no competing interests exist.”

2. We have noticed that you have uploaded supporting information but you have not included a list of legends. Please add a full list of legends for all supporting information files (including figures, table and data files) after the references list.

Additional Editor Comments (if provided):

Thanks for your thoughtful response to the comments from our reviewers. The study offers helpful insights into the important question of model generalizability. I have one minor request/comment. The significant skew in the data might be a bit of an issue. Ideally, analyses might include either upsampling or downsampling. Discussion of the potential impact of the skew - either in the discussion or as a limitation - would strengthen this paper.
---

## [Editor Report · Decision Letter 2]

17 Feb 2022

Generalizability Challenges of Mortality Risk Prediction Models: A Retrospective Analysis on a Multi-center Database

PDIG-D-21-00052R2

Dear Mr. Singh,

We are pleased to inform you that your manuscript 'Generalizability Challenges of Mortality Risk Prediction Models: A Retrospective Analysis on a Multi-center Database' has been provisionally accepted for publication in PLOS Digital Health.

Best regards,

Harry Hochheiser

Section Editor

PLOS Digital Health